# Polyamine Depletion Strategies in Cancer: Remodeling the Tumor Immune Microenvironment to Enhance Anti-Tumor Responses

**DOI:** 10.3390/medsci10020031

**Published:** 2022-06-10

**Authors:** Alexander Chin, Charles J. Bieberich, Tracy Murray Stewart, Robert A. Casero

**Affiliations:** 1Department of Biological Sciences, University of Maryland, Baltimore County, Baltimore, MD 21250, USA; achin2@umbc.edu (A.C.); bieberic@umbc.edu (C.J.B.); 2University of Maryland Marlene and Stewart Greenebaum Cancer Center, Baltimore, MD 21201, USA; 3Department of Oncology, Sidney Kimmel Comprehensive Cancer Center, Johns Hopkins University School of Medicine, Baltimore, MD 21287, USA; tmurray2@jhmi.edu

**Keywords:** polyamines, tumor microenvironment, cancer therapeutic, immune regulation, difluoromethylornithine, polyamine blocking therapy, macrophage polarization

## Abstract

Polyamine biosynthesis is frequently dysregulated in cancers, and enhanced flux increases intracellular polyamines necessary for promoting cell growth, proliferation, and function. Polyamine depletion strategies demonstrate efficacy in reducing tumor growth and increasing survival in animal models of cancer; however, mechanistically, the cell-intrinsic and cell-extrinsic alterations within the tumor microenvironment underlying positive treatment outcomes are not well understood. Recently, investigators have demonstrated that co-targeting polyamine biosynthesis and transport alters the immune landscape. Although the polyamine synthesis-targeting drug 2-difluoromethylornithine (DFMO) is well tolerated in humans and is FDA-approved for African trypanosomiasis, its clinical benefit in treating established cancers has not yet been fully realized; however, combination therapies targeting compensatory mechanisms have shown tolerability and efficacy in animal models and are currently being tested in clinical trials. As demonstrated in pre-clinical models, polyamine blocking therapy (PBT) reduces immunosuppression in the tumor microenvironment and enhances the therapeutic efficacy of immune checkpoint blockade (ICB). Thus, DFMO may sensitize tumors to other therapeutics, including immunotherapies and chemotherapies.

## 1. Introduction

The major polyamines found in mammals, putrescine, spermidine, and spermine, are polycationic alkylamines that are essential for life [1,2]. Polyamines perform critical functions, including modulation of cell growth, proliferation, and function. The biosynthetic and catabolic machinery are highly regulated; dysregulation can promote tumorigenesis and/or oxidative damage and cell death [1,2,3]. The biosynthetic rate-limiting enzyme ornithine decarboxylase (ODC) catalyzes the generation of the first polyamine putrescine (Figure 1). A second rate-limiting enzyme, S-adenosylmethionine decarboxylase (AMD1), produces the aminopropyl donor sequentially added by spermidine synthase and spermine synthase to generate spermidine and spermine, respectively. As the major catabolic enzymes, spermidine/spermine N1-acetyltransferase (SAT1) acetylates spermidine or spermine; spermine oxidase (SMOX) directly oxidizes spermine to generate hydrogen peroxide, spermidine, and 3-aminopropanal; and N1-acetylpolyamine oxidase (PAOX) is a peroxisomal enzyme that can oxidize the acetylated polyamines, producing 3-aceto-aminopropanal and either spermidine or putrescine, depending on the starting substrate; however, most acetylated polyamines are exported from the cell via solute carriers such as SLC3A2 [4]. Recently ATP13A3, a P-type ATPase, has been implicated as a major polyamine importer in mammalian cells [5,6]; however, the precise polyamine transport machinery and mechanisms remain unclear.

Overexpression of ODC has been implicated in numerous cancers [9], while genetic deletion of *Odc1* is embryonic lethal in mice [10]. The proto-oncogene *MYC* is highly overexpressed and/or amplified in human cancers and transcriptionally activates *ODC1*; thus, polyamine metabolism is an attractive cancer therapeutic target [1,11]. The most widely used inhibitor of ODC is 2-difluoromethylornithine (DFMO), an enzyme-activated suicide inhibitor that is well tolerated in humans and animal models. DFMO is FDA-approved for African trypanosomiasis but not for cancer due to its limited efficacy as a single cancer agent [1]; however, combination treatments targeting additional pathways engaged in polyamine metabolism have shown efficacy in animal models and are being tested in clinical trials [2,9,12]. Currently, there are four clinical trials listed (clinicaltrials.gov) investigating DFMO with chemotherapeutics in astrocytoma (NCT02796261, active) and neuroblastoma (NCT03794349, recruiting; NCT02030964, active; NCT01059071, completed). Additionally, one trial is investigating DFMO with the polyamine transport inhibitor AMXT 1501 in solid tumors (NCT03536728, recruiting), and another is investigating the polyamine analog SBP-101 with gemcitabine in pancreatic cancer (NCT03412799, completed). Of note, only one trial has results posted (NCT01059071, Phase I) in which DFMO and etoposide co-treatment was well-tolerated and resulted in very low frequencies of serious adverse events.

Analysis of the immunomodulatory effects of polyamine depletion strategies is the focus of this review. Specifically, new evidence demonstrates the requirement of polyamines for regulating immune cell lineage, proliferation, and function. Thus, systemic inhibition of biosynthetic and metabolic machinery may alter the immune microenvironment and the balance between pro- and anti-inflammatory networks. Combined polyamine depletion strategies effectively reduce tumor burden and increase survival, but the treatment effects on the immune microenvironment are just beginning to be investigated.

## 2. New Evidence Indicates That Polyamines Are Critical for Immune Cell Specification and Function

### 2.1. Deficiency in Polyamine Biosynthesis in CD4+ T Cells Confers Aberrant Function and Delayed Proliferation but Enhances IFNγ Expression

CD4+ T cells require polyamines for proper lineage specification and function. To specifically delete *Odc* in CD4+ T cells, Puleston et al. bred mice harboring loxP sites flanking exons 9–11 of *Odc* with mice expressing CD4^cre^ to generate *Odc-∆T* mice on the C57BL/6 (B6) background [13]. Similar to WT B6 mice, *Odc-∆T* mice had comparable CD4+ and CD8+ T cell frequencies and numbers in the spleen; however, deletion of *Odc* in CD4+ T cells led to aberrant differentiation and cytokine profiles.

Naïve CD4+ T cells were isolated from WT and *Odc-∆T* mice, and the T cells were induced to differentiate into Th1, Th2, Th17, and Treg subsets in vitro [13]. From day 0 to 4, all differentiated T-helper cells derived from *Odc-∆T* mice exhibited delayed proliferation as well as an approximately 4-fold decrease in cell number as compared to those of WT mice. Examining cytokine profiles revealed critical differences. Namely, differentiated Th2, Th17, and Tregs exhibited significantly increased IFNγ expression (from negligible levels to between 40–60% cells expressing IFNγ). Interestingly, Th17 cells exhibited near-complete loss of IL-17A expression while Th1, Th2, and Tregs gained expression of IL-17A. Moreover, IFNγ expression was sustained across multiple cell divisions.

To examine if enhanced IFNγ expression and inflammation could be induced in vivo, naïve CD4+ T cells from WT or *Odc-∆T* mice were transferred into lymphocyte-deficient *Rag1^−/−^* mice. After 23 days, *Rag1^−/−^* mice bearing *Odc^−/−^* T cells began losing weight. These mice developed severe inflammation (physiologic and histologic), and harbored a significantly increased number of IFNγ+ T cells, and by day 36 had to be sacrificed due to a loss of 20% of their body weight compared to *Rag1^−/−^* mice bearing WT T cells [13].

Recent evidence has implicated polyamines in the post-translational modification of eukaryotic translation initiation factor 5A (eIF5A) as a critical juncture through which polyamines mediate cellular processes [14,15]. eIF5A is the only protein known to be post-translationally modified via hypusination: (1) deoxyhypusine synthase (DHPS) transfers the 4-aminobutyl group derived from spermidine to the eIF5A^Lys50-deoxyhypusine^; (2) deoxyhypusine hydroxylase (DOHH) catalyzes the hydroxylation to generate eIF5A^hyp^. Since global deletion of *Dhps* is embryonic lethal in mice, cell-type-specific deletions are required [16]. Puleston et al. generated *Dohh-∆T* and *Dhps-∆T* mice to specifically delete DOHH and DHPS, respectively, in CD4+ T cells [13]. In the spleen of *Dohh-∆T* mice, there was an approximately 50% reduction in T cell frequency and number as compared to that in WT mice. Similar to *Odc* deletion, in vitro differentiated *Dohh^−/−^* and *Dhps^−/−^* CD4+ T cells exhibited highly elevated IFNγ and IL-17A expression. Importantly, *Dohh-∆T* mice died within 2–3 months, likely from severe inflammation based on physiological and histological analysis. Furthermore, aberrant *Odc^−/−^* and *Dohh^−/−^* Tregs failed to protect against the heightened inflammatory response as compared to WT Tregs.

Further mechanistic analyses revealed that *Odc^−/−^* and *Dohh^−/−^* T cells harbored aberrant acetylation marks. Pharmacological inhibition and/or genetic ablation of the H3K27 acetyltransferase P300 or the H3K9 acetyltransferase KAT2A reduced IFNγ levels across Th subsets to WT levels. Thus, depletion of hypusine biosynthesis recapitulates the consequence of *Odc* deletion, and ultimately suggests that polyamine-derived hypusine is critical for specifying T-helper cell lineage.

### 2.2. Spermidine Potentiates Foxp3-Expressing Tregs

In an independent study, spermidine was also shown to regulate T-cell differentiation [17]. Specifically, spermidine induced Foxp3 expression in Th17 and Treg cells generated by in vitro differentiation of murine naïve CD4+ T cells. Interestingly, naïve CD4+ T cells derived from human cord blood samples induced to differentiate under Th17 conditions and supplemented with spermidine also reduced IL-17 production. To investigate spermidine supplementation in vivo, Carriche et al. treated BALB/c mice with 30 mM spermidine in drinking water and the adoptive transfer of naïve T cells [17]. After 1 week of treatment, there was a significant increase in Foxp3+ Treg cells within the small intestine and colon. Furthermore, L-arginine supplementation in drinking water also led to increased CD4+ Foxp3+ T cells in the colon. Moreover, the deletion of autophagy protein 5 (ATG5) in T cells abolished the increase in Foxp3+ Tregs. Thus, autophagy may play an important role in mediating spermidine potentiation of Tregs. To measure the immunosuppressive Treg functions of spermidine-treated mice, naive CD4+ T cells were transferred into lymphocyte-deficient *Rag2^−/−^* mice, and 30 mM spermidine was administered in drinking water. Importantly, spermidine treatment ameliorated the severe body weight loss observed in the untreated group. Moreover, there was a significant increase in frequency and number of Foxp3+ T cells in the colon and small intestine, concomitant with increased anti-inflammatory IL-10 production [17].

### 2.3. Polyamine Transport Can Compensate for Deficiency in Biosynthesis in CD4+ T Cells

In agreement with findings from Puleston et al. [13], Wu et al. noted that *Odc^−/−^* T cells exhibited delayed cell cycle progression post-activation, and overall suppressed proliferation in vitro [18]. Pharmacological inhibition of ODC via DFMO treatment similarly reduced T cell progression and proliferation in vitro. Moreover, genetic deletion or pharmacological inhibition resulted in increased T cell death. To examine the consequences of *Odc* deletion in vivo, CD4+ T cells from *Odc-∆T* or WT mice were transferred to *Rag1^−/−^* mice. Interestingly, loss of ODC neither significantly affected T cell proliferation nor antigen-specific proliferation. Wu et al. commented that polyamine salvage from circulation might be responsible for supporting T cell proliferation and function in vivo to compensate for the deficiency in de novo polyamine biosynthesis [18]. This is an interesting perspective that agrees with comments by Carriche et al. that secreted polyamines in the microenvironment and/or serum may promote distribution and uptake by surrounding cells. Furthermore, Carriche et al. posit that the gut microbiome may serve as a major source of circulating polyamines [17].

To investigate if exogenous polyamine uptake can mediate T cell proliferation in the absence of ODC, the authors supplemented polyamines and observed restored cell cycle progression, proliferation, and viability in *Odc^−/−^* CD4+ T cells in vitro [18]. Moreover, the combination treatment of DFMO and the polyamine transport inhibitor AMXT 1501 (AMXT) abolished the compensatory increase in putrescine uptake observed in DFMO treatment alone. Similar results were obtained with human T cells. In an experimental autoimmune encephalomyelitis (EAE) murine model of multiple sclerosis (MS), DFMO and AMXT abrogated pathogenesis, which the authors attributed to the reduction in CD4+ T cell infiltration and in IL-17+ CD4+ T cells; however, the disease state induces hyper-recruitment of a complex array of immune cells [18,19]. Nonetheless, polyamine biosynthesis and transport can effectively maintain polyamine homeostasis in and viability of CD4+ T cells.

Surprisingly, Wu et al. showed that *Odc* deletion in CD4+ T cells, which were differentiated under Th1, Th17, and Treg conditions, resulted in significantly reduced populations of IFNγ+, IL-17+, as well as increased Foxp3+ T cells [18]. This is in sharp contrast to findings by Puleston et al., in which *Odc* deletion in CD4+ T cells cultured under similar conditions resulted in significantly increased IFNγ+, IL-17+, and Foxp3+ T cells. Moreover, the differences were highly significant. Of note, however, while Foxp3 expression by Th1 and Th2 was highly significant (*p* < 0.00005), the change in percent value of expressing cells was very small, perhaps indicative of a switch from non-expressing to expressing cells. Nonetheless, the data demonstrate opposing effects of *Odc* deletion. Although Puleston et al. cite the work by Wu et al., the authors do not mention these important differences. The cause for such differences is unclear but may involve differences in *Odc-∆T* model generation (CD4^cre^ is constant), and/or differences within in vitro differentiation and flow cytometric analyses; the disparities warrant further study.

### 2.4. Odc Deletion in Myeloid Cells Enhances Pro-Inflammatory Response to Bacterial Infection

In the context of bacterial infection, the potentiation of classically activated M1 macrophages contributes to a pro-inflammatory response [20]. To investigate the role of myeloid cell ODC in response to *H. pylori* or *C. rodentium* infections, Hardbower et al. generated *Odc^∆mye^* mice (*Lyz2-Cre; Odc^loxp/loxp^*). *H. pylori* or *C. rodentium* infection enhanced acute and chronic inflammation in *Odc^∆mye^* mice [21]. Gastric tissue from infected *Odc^∆mye^* mice harbored significantly increased inflammatory cytokines and chemokines as compared to that of control mice (*Odc^loxp/loxp^*). Moreover, gene expression data revealed increased expression of classical M1 markers such as *Il1b*, *Il6*, *Tnfa*, *Nos2*, and limited effect on M2 markers such as *Arg1*, *Chil3*, *Retnla*, and *Il10* [21].

To confirm the specific engagement of macrophages, the authors isolated bone marrow-derived macrophages (BMDMs) from *Odc^∆mye^* and control mice and infected them with either *H. pylori* or *C. rodentium*. In agreement with their findings, mRNA expression of M1 markers, as well as secretion of inflammatory cytokines, were significantly increased in *Odc^−/−^* BMDMs. The authors then examined epigenetic alterations and observed that *Odc^−/−^* BMDMs had increased the presence of H3K9ac in promoters of M1 markers. Subsequent pharmacologic inhibition of KAT2A significantly reduced M1 marker expression while exogenous putrescine reverted levels of NO production as well as histone modifications [21]. Interestingly, KAT2A inhibition also decreased pro-inflammatory markers of *Odc^−/−^* T-helper cells [13]; thus, Odc deletion may exert conserved epigenetic alterations across immune populations.

### 2.5. Microbiota-Derived Polyamines Are Important for Modulating Macrophage Function

Exogenous polyamines can be imported into cells and promote biosynthetic and metabolic activity. Recent evidence has highlighted the importance of microbiota-derived polyamines in regulating macrophage differentiation in vivo [22]. Germ-free B6 mice were inoculated with either WT *E. coli* or SK930 *E. coli*, a putrescine biosynthesis-deficient strain, and fecal and colonic tissue samples were analyzed. Interestingly, mice harboring SK930 *E. coli* had significantly reduced cell number and percent of anti-inflammatory CX3CR1^hi^ Ly6C- M2-like macrophages in the colonic lamina propria as compared to that of mice with WT *E. coli* [22]. Moreover, pro-inflammatory Nos2+ Arg1- F4/80+ CD11b+ macrophages were significantly increased. Thus, bacterium-derived polyamines may potentiate anti-inflammatory macrophages, similar to how Carriche et al. reported an increase in immunosuppressive Tregs upon spermidine administration in drinking water [17].

### 2.6. The Role of Hypusine Biosynthesis in Specifying Macrophage Polarization Is Unclear

Recent evidence has also implicated hypusine in the modulation of macrophage activation. BMDMs were isolated from B6 mice and polarized into classically activated M1 (LPS and IFNγ) macrophages and alternatively activated M2 (IL-4) macrophages [23]. Upon M2 induction, the authors observed increased ODC, DHPS, and eIF5A^hyp^ levels as well as increased putrescine uptake as compared to that of M1-induced and unpolarized BMDMs. Another group also observed increased levels of hypusine as well as mitochondrial electron-transport complexes upon M2 induction as compared to M1 induction [22]. Moreover, Puleston et al. reported that pharmacological inhibition of DHPS via *N*^1^-guanyl-1,7-diaminoheptane (GC7) significantly reduced M2 markers in IL-4 stimulated human monocyte-derived macrophages [23]. In vivo injection of an IL-4 complex significantly increased levels of eIF5A^hyp^, which the authors attributed to the potentiation and accumulation of Ly6G- Ly6C- F4/80+ CD11b+ M2 macrophages [22,24,25].

In contrast to findings by Puleston et al. and Nakamura et al., neither IL-4 induction of M2 macrophages affected eIF5A^hyp^ levels, nor did pharmacological inhibition of DHPS via GC7 affect eIF5A^hyp^ levels [16]. The authors noted this disparity in their results and suggested that GC7 had an effect on M2 polarization independent of DHPS inhibition. To further investigate the role of DHPS in macrophage activation, Anderson-Baucum et al. generated *Dhps^Δmye^* mice harboring a genetic deletion of *Dhps* in the myeloid cell population (*Lyz2-Cre*) [16]. Interestingly, BMDMs derived from *Dhps^Δmye^* mice and polarized under M1 conditions did not differ in their secretion of inflammatory cytokines TNFα and IL-6 or chemokines MCP-1 and MIP-2 as compared to that of control mice. Proteomic analyses in M2-polarized macrophages derived from *Dhps^Δmye^* mice apparently did not yield clear pathway alterations in macrophage functions; thus, the authors concluded that DHPS-deficiency primarily affects M1 polarization [16]. Of note, while the relative abundance of DHPS was significantly decreased compared to control, it was still relatively high. The disparities in these results warrant further study.

### 2.7. Immune Cell-Intrinsic Deficiencies in Polyamine Biosynthesis Induced by Odc Deletion or Pharmacological Inhibition via DFMO Can Restrict Proliferation and Alter Function

It is evident that polyamines and hypusine also promote the proper function of other immune cells, such as NK cells. Pharmacological modulation of polyamine biosynthesis via DFMO or *N*^1^,*N*^11^-diethylnorspermine (DENSpm) reduced NK cell proliferation as well as IFNγ and granzyme B production in vitro [26]. Moreover, inhibition of hypusination via GC7 similarly reduced IFNγ and granzyme B production as well as cytotoxicity, without affecting viability. Thus, decreased polyamines, especially spermidine, are detrimental to NK cell function in vitro [26], bringing to question the effects of systemic polyamine depletion strategies in disease settings such as cancer.

The data presented on T cells, macrophages, and NK cells thus far exemplify the cell-intrinsic immunomodulatory effects of polyamine depletion. Since polyamines are critical for regulating immune cell lineage, proliferation, and function, it is expected that systemic polyamine depletion strategies would affect immune populations. Interestingly, while genetic deletion or pharmacological inhibition of ODC in CD4+ T cells, macrophages, and NK cells led to reduced proliferation and/or increased cell death in vitro, the aberrant CD4+ T cells (as well as Tregs) or macrophages presented higher IFNγ expression that was perhaps indicative of a pro-inflammatory phenotype [13,21,26]. On the other hand, spermidine potentiated Tregs in vitro and/or stimulated an anti-inflammatory phenotype in vivo [17,22]. Thus, depletion of polyamines may tip the balance between pro- and anti-inflammatory signaling toward pro-inflammatory even though absolute immune content may decrease, at least in vitro. In vivo, serum-available polyamines derived from sources such as the gut microbiome may compensate for disrupted de novo biosynthesis and maintain immune cell proliferation [17,18,22].

The addition of a tumor further confounds the immune microenvironment. High intratumoral ARG1 expression is associated with an immunosuppressive phenotype, which may deplete resources for polyamine biosynthesis of immune cells such as T cells. Concomitantly, tumor-derived polyamines may potentiate Tregs and M2 macrophages. Furthermore, tumor cell apoptosis and subsequent efferocytosis by neighboring and/or infiltrating immune cells may enhance the immunosuppressive phenotype. McCubbrey et al. demonstrated that LPS-stimulated macrophages co-cultured with apoptotic Jurkat cells exhibited increased spermine uptake and reduced *IL1β* and *IL6* gene expression, while treatment with the Rac-1 inhibitor NSC23766 abolished the increase in polyamine content and reverted *IL1β* and *IL6* expression to near-normal levels [7]. Thus, investigating immune-cell-intrinsic alterations upon systemic DFMO treatment alone and/or in combination with transport inhibitors may provide insight into the anti-tumor efficacy or lack thereof in these treatment groups.

## 3. Immunomodulatory Effect of DFMO in the Context of Polyamine Blocking Therapy

### 3.1. DFMO and AMXT 1501 Reduce Tumor Growth in Immunocompetent Mice

Polyamine-blocking therapy (PBT) combining DFMO (0.5% *w*/*v*, drinking water) and AMXT 1501 (3 mg/kg, i.p.) significantly reduced putrescine and spermidine levels and inhibited tumor growth, as compared to either treatment alone, in syngeneic models of colon carcinoma (CT26.CL25, BALB/c) and melanoma (B16F10, B6) [27]. Interestingly, nude mice (deficient in mature T cells and reduced lymphocytes) injected with CT26.CL25 cells did not respond to PBT. These data suggest that the PBT-mediated anti-tumor response was dependent on T cells. Furthermore, PBT prevented tumor rechallenge in a syngeneic model of mammary adenocarcinoma (Neu02, FVB), suggesting that PBT promotes a protective anti-tumor immune memory [27]. Thus, while PBT-induced immunosurveillance is likely highly context-dependent, these data provide a strong rationale for investigating the therapeutic benefit of PBT and subsequent alterations in the tumor immune microenvironment.

### 3.2. DFMO Alone Can Reduce Immunosuppression in the Tumor Microenvironment

In the context of glioblastoma, infiltration of immunosuppressive tumor-associated myeloid cells (TAMCs) is associated with poor survival [28]. Intracranial injection of CT-2A glioma cells into B6 mice yielded aggressive tumors with significantly increased polyamine content and ODC expression [28]. The authors treated CT-2A tumor-bearing mice with DFMO (1% *w*/*v*) and observed increased survival and reduced polyamines in TAMCs but not in myeloid cells within the spleen. To investigate potential immune-associated effects of DFMO treatment, the authors injected *Rag1^−/−^* mice (lacking T and B cells) with CT-2A glioma cells and observed no survival benefit with DFMO. Thus, lymphocytes could have directed the PBT-mediated anti-tumor response. Interestingly, in B6 mice, the authors observed significant reductions in M-MDSCs, TAMs, and microglia but not in the T cell compartment. Thus, the active immunosurveillance driving a survival benefit may differ depending on tumor context and model system. Indeed, in the GL-261 syngeneic model, treatment with DFMO led to a reduction in TAMCs but also in CD4+ T cells. Nonetheless, reduction in TAMC populations was a common effect in mice exhibiting survival benefits. The authors further demonstrated that, in vitro, TAMCs tolerated increasingly acidic conditions (down to pH 6.7), but the addition of DFMO significantly enhanced necrosis, to which add-back of putrescine was able to rescue. These data suggest that polyamines buffer intracellular pH and might protect TAMCs against the acidic TME in glioblastoma [28].

Importantly, CT-2A tumors presented higher immune-inhibiting programmed cell death-ligand 1 (PD-L1) expression upon DFMO treatment, to which putrescine reverted to near-control levels. To leverage this outcome, the authors combined DFMO with either immune-stimulating checkpoint inhibitor α-PD-L1 or α-PD-1 and observed significantly increased survival (additive effect) as compared to single treatment arms [28]. Thus, DFMO alone may reduce the immunosuppressive TME and sensitize the tumor to immune checkpoint blockade (ICB).

### 3.3. DFMO and Trimer44NMe Reduce Immunosuppression and Sensitize Tumors to α-PD-1

Muth et al. delineated the development of a novel polyamine transport inhibitor (PTI), Trimer44NMe, designed/optimized to inhibit spermidine import in DFMO-treated Chinese hamster ovary (CHO) and L3.6pl human pancreatic cancer cells in vitro [29]. To investigate the efficacy of this combination in vivo, Alexander et al. subcutaneously injected female B6 mice with 5 × 10^5^ B16F10-sTAC melanoma cells and administered DFMO (0.25% *w*/*v*, drinking water) and/or the Trimer PTI (3 mg/kg, i.p. daily) [30]. Trimer PTI preferentially bound to the tumor but alone did not affect tumor growth; however, the combination of DFMO and Trimer PTI (PBT) significantly reduced tumor volume and polyamine content. Moreover, there was a significant increase in tumoral F4/80+ cells as well as in IFNγ, IL-10, and MCP-1 expression, indicative of immune infiltration [30].

To confirm PBT efficacy in vivo, the authors subcutaneously injected 5 × 10^5^ CT26.CL25 colon cancer cells into BALB/c mice and treated them with PBT. Analysis of tumor-isolated leukocytes revealed a significant increase in CD8+ T cells with increased IFNγ and granzyme B expression as well as decreases in CD206+ F4/80+ macrophages, Gr1+ CD11b+ MDSCs, and CD25+ CD4+ Tregs. Unfortunately, DFMO and PTI were not examined as single agents in this system. To further examine the role of T cells, the authors depleted CD-4 and CD-8-expressing cells (75 μg anti-CD4/CD8, i.p. every three days, four doses total) and observed accelerated tumor growth; subsequent DFMO and Trimer PTI co-treatment did not significantly reduce tumor growth [30]. Thus, cytotoxic and helper T cell populations may drive the anti-tumor response. Interestingly, although combined DFMO (1% *w*/*v*) and Trimer PTI (1.8 mg/kg) conferred significant survival benefits in the orthotopic Pan02 model of gemcitabine-resistant pancreatic cancer, it did not further reduce tumor burden when compared to DFMO or PTI alone [31]. Nakkina et al. increased the Trimer44NMe dosage to 4 mg/kg, (DFMO: 0.25% *w*/*v*) and observed a significant increase in survival, reduction in Pan02 pancreatic tumor burden, and increase in co-stimulatory marker CD86 expression when compared to either treatment alone [32].

To investigate PBT-driven alterations of immune populations, Alexander et al. expanded their previous findings to include an orthotopic model of breast cancer. The 4T1 cells were injected into the mammary fat pad of BALB/c mice and were treated with previously established regimens [30]. Co-treatment with DFMO and Trimer PTI significantly inhibited tumor growth as well as reduced metastatic lung nodules as compared to untreated mice. Tumor progression in the 4T1 model is associated with high recruitment and accumulation of MDSCs [33]. Analysis of tumor composition revealed 60% leukocytes, which was significantly reduced by nearly 20% in PBT-treated mice. PBT reduced Ly6G+ CD11b+ granulocytic MDSCs and CD206+ F4/80+ macrophages. Interestingly, treatment with anti-Ly6G prior to PBT abolished the inhibition of tumor growth, suggesting that Ly6G+ myeloid cells were targeted in PBT [33]. Of note, Boivin et al. investigated the duration and efficacy of anti-Ly6G depletion of Ly6G+ cells and observed a re-emergence of neutrophilic cells [34]. The authors proposed a continuous dosing regimen to reliably deplete neutrophils [34]. Thus, whether neutrophils and/or other Ly6G-expressing cells remained after the initial dosing remains unclear.

To leverage the decrease in TME immunosuppression, Alexander et al. tested PBT in combination with α-PD-1 in the 4T1 and B16F10-sTAC tumor models [33]. Although PBT alone resulted in reduced tumor growth and improved survival, the addition of α-PD-1 yielded additional benefits. Interestingly, α-PD-1 treatment alone had no effect on inhibiting tumor growth in the 4T1 model, unlike in the B16F10-sTAC model. A T-cell response may be critical for PBT-mediated inhibition of tumor growth in melanoma and colon carcinoma; however, depletion of pro-tumorigenic monocytic and/or granulocytic myeloid populations may be critical for positive treatment outcomes in mammary carcinoma [30,33]. Thus, the drivers of immune alterations and/or response are tumor- and context-dependent.

### 3.4. DFMO and 5-Azacytidine Reverse Immunosuppression and Enhance Pro-Inflammatory Myeloid-Derived Cells

Previous studies have demonstrated that epigenetic therapies such as DNA methyltransferase inhibitors (DNMTIs) and histone deacetylase inhibitors (HDACIs) can reduce tumor immunosuppression [35]. In the VEGF-β-Defensin ID8 (VDID8) murine ovarian cancer model, the combination of the DNMTI 5-azacytidine (5AZA-C) with the HDACI MS275 enhanced IFN signaling and sensitized tumors to α-PD-1 immunotherapy [35].

To investigate the potential for combining epigenetic therapy with polyamine depletion, Travers et al. treated VDID8 tumor-bearing B6 mice with DFMO (2% *w*/*v*) and/or 5AZA-C (0.5 mg/kg, i.p.) [36]. DFMO or 5AZA-C alone or in combination increased lymphocytes in ascites and particularly IFNγ+ cells, including NK cells, as well as T cells. Combination treatment significantly reduced tumor ascites volume and increased survival as compared to either treatment alone, as well as reduced the CD11b+ F4/80+ macrophage compartment. Interestingly, the addition of α-PD-1 neither decreased tumor burden nor increased survival, suggesting that the T-cell response was not the primary mechanism.

To investigate macrophage polarization, the authors treated mice continuously (biweekly) with anti-colony stimulating factor 1 receptor (α-CSF1R) and observed a significant reduction in MHC II+ CD206- M1 macrophages while MHC II- CD206+ M2 macrophages remained low. Moreover, α-CSF1R treatment increased tumor ascites and reduced survival of mice treated with DFMO and 5AZA-C; thus, the authors proposed that M1 macrophages were critical players contributing to the positive treatment outcome [36]. The limited therapeutic effect of α-CSF1R-mediated depletion of TAMs in various tumors has been documented [37,38]. Kumar et al. demonstrated that inhibition of CSF1R induced PMN-MDSC infiltration [37]; thus, while a reduction in the M1 macrophage compartment observed by Travers et al. likely contributed to the increased tumor burden, it is possible that additional factors were involved, such as the recruitment of MDSCs, that was not sustainably affected by DFMO plus 5AZA-C treatment.

Notably, the tumors examined thus far exhibit relatively high frequencies of tumor-associated immune cells. Polyamine depletion strategies may have an increased efficacy in such tumors (Figure 2); however, there exists a paucity of these studies in immunologically deprived solid tumors. As described by Gitto et al., PBT with DFMO and Trimer PTI did not yield additive survival benefits as compared to DFMO alone in a gemcitabine-resistant pancreatic adenocarcinoma model [31]. Furthermore, the model system used by Travers et al. utilizes ascites volume as a measure of tumor burden as well as response to treatment [36]; however, the solid ovarian tumor microenvironment likely differs in immune content and treatment-induced response. It would be interesting to examine both therapeutic efficacy and/or immunomodulatory effects of polyamine depletion strategies within poorly immune-infiltrated solid tumors.

## 4. Conclusions and Future Directions

Combining polyamine depletion strategies to restrict cancer cell proliferation has shown efficacy within in vitro systems and in vivo animal models and is currently being tested in clinical trials [2,9,12]. Recent evidence has demonstrated that the tumor immune microenvironment is significantly altered, that the efficacy of PBT may be due to the treatment-induced anti-tumor immune response, and that the immune cell subtypes driving this response differ depending on tumor model systems [27,28,30,33,36]. Thus, it is of interest to define the immunomodulatory effects of these novel therapies. Importantly, new evidence in pre-clinical models of inflammation demonstrates the critical regulatory effects exerted by polyamines on immune cell lineage specification, proliferation, and function. In the tumor context, these data may provide rationale for targeting polyamine depletion strategies toward the tumor and/or immediate pro-tumorigenic immune cells as a means to increase the efficacy of DFMO alone and/or in combination with novel therapeutics.

The mouse models tested thus far have exemplified the PBT-mediated remodeling of the immunological landscape, primarily promoting a pro-inflammatory and/or anti-tumor response; however, the location of tumor implantation affects the immune microenvironment. In a comparison of mouse models of colorectal cancer, Zhao et al. developed a novel endoscopy-guided orthotopic model and compared the immune content of it and other syngeneic orthotopic models to that in respective subcutaneous models [39]. Notably, orthotopic tumors harbored significantly higher immune infiltration of T cells, B cells, and NK cells, as well as fewer immunosuppressive myeloid cells than subcutaneous tumors, with increased expression of inflammatory cytokines IL-2, IL-6, and IFNγ. Interestingly, while subcutaneous tumors exhibited higher levels of immune-inhibiting checkpoint cytotoxic T-lymphocyte associated protein 4 (CTLA-4) and PD-1 on T cells as compared to orthotopic tumors, these tumors persisted post α-PD-1 and α-CTLA-4 treatment while orthotopic tumor growth ceased [39]. Thus, the immune landscape associated with carcinogenesis differs significantly with tumor location and can translate into differential treatment response. In the studies utilizing subcutaneous models, investigating the effects of combined polyamine depletion strategies on orthotopic and/or transgenic models would provide critical data for the characterization and stratification of tumor immune microenvironments.

As recent evidence demonstrated that *Odc* deletion or DFMO treatment resulted in reduced proliferation and viability of T cells and NK cells cultured in vitro [18,26], systemic polyamine depletion might attenuate the immune infiltrate in vivo. Thus, it would be of interest to develop and investigate tumor-specific polyamine depletion strategies. One approach may involve nanoparticle encapsulation of, or conjugation with, DFMO and other polyamine analogs to deliver and accumulate the drugs within the tumor [2]; however, Wu et al. demonstrated that *Odc* deletion in T cells did not affect proliferation in vivo, likely due to compensatory mechanisms [18]; perhaps systemic DFMO would not significantly inhibit immune proliferation. Nonetheless, developing tumor-targeted delivery mechanisms with DFMO could clarify whether the anti-proliferative effects on immune populations and/or mechanistic switch to pro-inflammatory phenotypes are important in determining treatment outcomes. Moreover, while in vitro generated macrophages and T cells polarized under established conditions provide a system to investigate polyamine depletion, the in vivo immune content and profiles vary with the microenvironment; thus, classical signatures of M1 or M2 macrophages, as well as T-helper subsets characterized in vitro, might not be directly translatable in vivo [40].

The expansion of multi-omic sequencing has greatly advanced knowledge of intricate tumor landscapes. Single-cell RNA sequencing (scRNA-seq) harnesses genome-wide transcriptional profiling with single-cell resolution and has enabled the identification and characterization of the immune landscape as well as developmental trajectories [41,42]. Alshetaiwi et al. defined an MDSC differentiation trajectory and expanded the conventional MDSC gene signature to include CD84^hi^ and junctional adhesion molecule-like protein (JAML); their data highlighted that the standard markers CD11b and Gr-1 were also expressed in B cells, T cells, neutrophils, and monocytes in mammary tumor-bearing mice [41]. Thus, without functional confirmation, the presence of conventional MDSC markers is likely insufficient. This phenomenon extends into other immune cells; thus, performing scRNA-seq on tumor and lymphoid organs of DFMO-treated and PBT-treated mice may more clearly define the heterogeneity of the TME and of individual cells within clusters. Alternatively, digital spatial profiling (DSP) enables highly specific and multiplex transcriptomic and/or proteomic analyses and provides critical geographic readout [43]. With DSP, it may be possible to identify critical regions and major drivers of immunosuppression that are altered by DFMO treatment or PBT. In addition, epigenetic analyses may provide important mechanistic information. Interestingly, *Odc* deletion in BMDMs and T-helper cells resulted in enhanced expression of pro-inflammatory markers, which subsequent pharmacological inhibition of KAT2A abrogated; thus, *Odc* deletion may exert conserved epigenetic modifications across immune populations [13,21]. These data warrant further investigation of polyamine depletion strategies and their cell-intrinsic and TME immunomodulatory effects from multi-omic approaches.

## Figures and Tables

**Figure 1 medsci-10-00031-f001:**
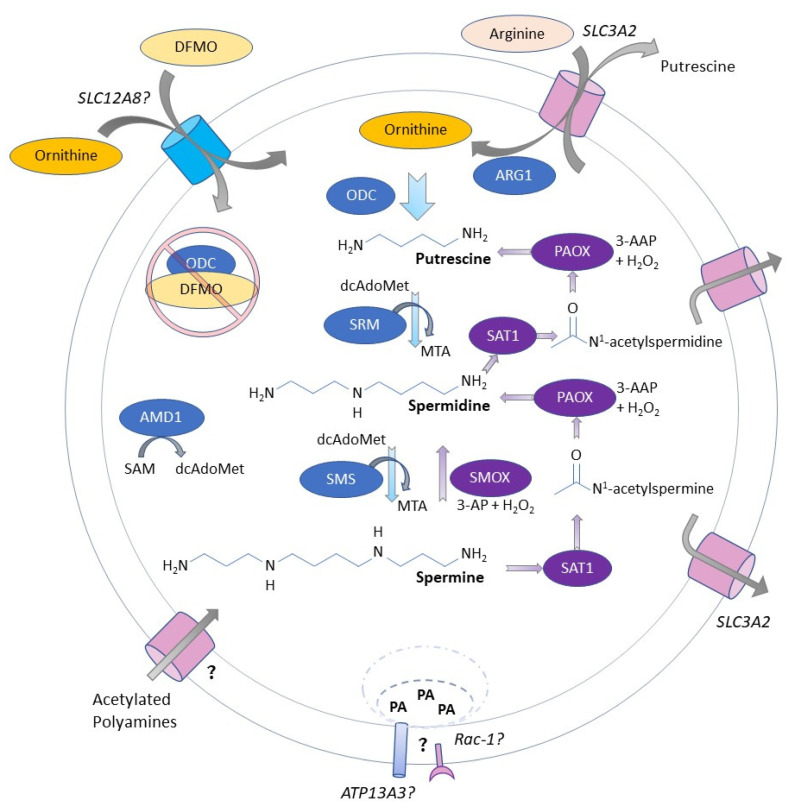
Diagram of polyamine metabolism. Ornithine is generated by the action of arginase I (ARG1) in the urea cycle. Ornithine decarboxylase (ODC) catalyzes the first rate-limiting reaction in polyamine biosynthesis to generate putrescine. Adenosylmethionine decarboxylase (AMD1) removes the carboxyl group from S-adenosylmethionine (SAM) to generate decarboxylated S-adenosylmethionine (dcAdoMet). Spermidine synthase (SRM) then adds an aminopropyl moiety from dcAdoMet to putrescine to generate spermidine. Spermine synthase (SMS) adds an additional aminopropyl moiety to spermidine to generate spermine. Polyamine biosynthetic enzymes are shown in blue. Spermine oxidase (SMOX), a cytoplasmic and nuclear oxidase, directly oxidizes spermine to spermidine and generates 3-aminopropanal (3-AP) and hydrogen peroxide (H_2_O_2_) as byproducts. Spermidine/spermine N1-acetyltransferase 1 (SAT1) adds an acetyl group to the N^1^ position of spermidine or spermine, thus permitting polyamine export as well as catabolism by polyamine oxidase (PAOX), a peroxisomal enzyme. PAOX generates spermidine or putrescine from the respective acetylated precursors, with 3-acetoaminopropanal (3-AAP) and H_2_O_2_ as byproducts. Polyamine catabolic enzymes are shown in purple. 2-Difluoromethylornithine (DFMO) is an ornithine analog that is imported into the cell and decarboxylated by ODC; the reaction intermediate covalently binds to and inactivates ODC. Import of polyamines (PA) is incompletely understood but may involve paired mechanisms engaging endocytosis [5,6,7,8].

**Figure 2 medsci-10-00031-f002:**
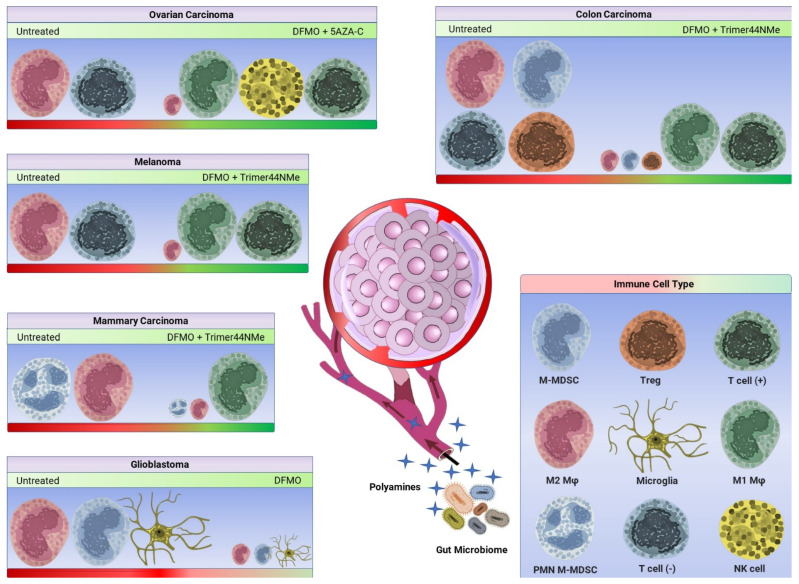
Polyamine blocking strategies decrease immunosuppression in multiple animal models of cancer. The tumor microenvironment (TME) presents anti-inflammatory cytokines and chemokines to support tumor growth; tumor- and gut microbiome-derived polyamines may further enhance immunosuppression [17,22]. Of the cancers examined, the respective panels are comprised of no-treatment (left) and treatment groups (right) and delineate significant alterations in the immune landscape. The red-green bar indicates the shift from an anti-inflammatory to pro-inflammatory phenotype, and the size of respective immune cells indicates the shift in observed frequencies as reported in the following articles. The immune cell types are labeled accordingly, and of note T cell (−) indicates a non-inflammatory and/or exhausted phenotype. DFMO and Trimer44NMe polyamine transport inhibitor (PTI) significantly reduced Ly6G+ polymorphonuclear myeloid-derived suppressor cells (PMN-MDSCs) and M2 macrophages (M2 Mφ) in the 4T1 orthotopic mammary carcinoma model [33]. DFMO and Trimer44NMe significantly increased IFNγ-expressing cells and T cells, and decreased M2 macrophages and/or monocytic MDSCs (M-MDSCs), as well as Tregs in the CT26.CL25 subcutaneous colon carcinoma model [30]. DFMO and Trimer44NMe significantly increased IFNγ-expressing myeloid cells and T cells in the B16F10-sTAC subcutaneous melanoma model [30,33]. DFMO treatment alone significantly decreased M2 Mφ and/or M-MDSCs as well as microglia in the CT-2A orthotopic glioblastoma model [28]. DFMO and 5-azacytidine (5AZA-C) significantly reduced M2 macrophages and increased pro-inflammatory M1 macrophages (M1 Mφ), T and NK cells in the VDID8 intraperitoneal ovarian carcinoma model [36].

## Data Availability

Data sharing not applicable.

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
