# Peer review of "Polyamine Depletion Strategies in Cancer: Remodeling the Tumor Immune Microenvironment to Enhance Anti-Tumor Responses"

_medsci, 2022, doi:10.3390/medsci10020031_

Round 1
Reviewer 1 Report
This manuscript entitled “Polyamine depletion strategies in cancer: remodeling the tumor immune microenvironment to enhance anti-tumor responses” provides a comprehensive and in-depth review on important roles of polyamine homeostasis in regulation of immune cell fate and tumor’s response to immunotherapy. The topics cover many important effects of polyamine biosynthesis on immune cell fate, immune microenvironment, inflammatory networks, and antitumor response. Importantly, new evidence suggests that targeting polyamine biosynthesis with potent inhibitors elicits antitumor immunity and enhances therapeutic efficacy of immune checkpoint inhibitors in poorly immunogenic tumors. These discoveries provide a strong rationale for combine polyamine depletion with immunotherapies to convert a “cold” non-immunogenic tumor into a “hot” one. The manuscript is very informative, up-to-date and covers remarkable development in polyamine field with comprehensive depth, balanced perspective, and intellectual insight.
Specific comments:
- Some recent studies showed that polyamine inhibition enhances infiltration of cytotoxic CD8+ T-cell into the tumor bed. This appears to be an important antitumor activity of polyamine depletion. It would be interesting to have some comments on this perspective.
- Polyamine depletion alone or in combination with other regimens, such as chemotherapy/immunotherapy, are being assessed in clinical trials (clinicaltrials.gov). The updated outcomes of these trials would help readers interpret the clinical significance.
- “Tumor-specific polyamine depletion strategy” was briefly discussed, but not quite clear how it works. Please add some clarity.
- What are the roles of polyamine catabolic enzymes-SSAT and SMOX in polyamine depletion-modulated changes of TME immune microenvironments?
Author Response
- Some recent studies showed that polyamine inhibition enhances infiltration of cytotoxic CD8+ T-cell into the tumor bed. This appears to be an important antitumor activity of polyamine depletion. It would be interesting to have some comments on this perspective.
We have made the following changes to delineate the appropriate cell populations:
“Thus, cytotoxic and helper T cell populations may drive the anti-tumor response.”
- Added to section 3.3 (revised lines 339-340)
“Recent evidence has demonstrated that the tumor immune microenvironment is significantly altered, that the efficacy of PBT may be due to the treatment-induced anti-tumor immune response, and that the immune cell subtypes driving this response differ depending on tumor model systems [26, 27, 29, 32, 35].”
- Added to section 4 (revised lines 449-451)
- Polyamine depletion alone or in combination with other regimens, such as chemotherapy/immunotherapy, are being assessed in clinical trials (clinicaltrials.gov). The updated outcomes of these trials would help readers interpret the clinical significance.
We thank the reviewer for this perspective and have made the following addition:
“Currently, there are four clinical trials listed (clinicaltrials.gov) investigating DFMO with chemotherapeutics in astrocytoma (NCT02796261, active) and neuroblastoma (NCT03794349, recruiting; NCT02030964, active; NCT01059071, completed). Additionally, one trial is investigating DFMO with the polyamine transport inhibitor AMXT 1501 in solid tumors (NCT03536728, recruiting), and another is investigating the polyamine analogue SBP-101 with gemcitabine in pancreatic cancer (NCT03412799, completed). Of note, only one trial has results posted (NCT01059071, Phase I) in which DFMO and etoposide co-treatment was well-tolerated and resulted in very low frequencies of serious adverse events.”
- Added to section 1 (revised lines 57-65)
- “Tumor-specific polyamine depletion strategy” was briefly discussed, but not quite clear how it works. Please add some clarity.
We have made the following changes to more clearly indicate that such strategies are very new, and to suggest an approach:
“to develop and to investigate tumor-specific polyamine depletion strategies”
- Modified in section 4 (revised line 480)
“One approach may involve nanoparticle encapsulation of, or conjugation with, DFMO and other polyamine analogues to deliver and accumulate the drugs within the tumor [2].”
- Added to section 4 (revised lines 481-482)
- What are the roles of polyamine catabolic enzymes-SSAT and SMOX in polyamine depletion-modulated changes of TME immune microenvironments?
Specific polyamine analogue treatment can induce either SSAT, SMOX, or both and lead to a decrease in the higher polyamines in the tumor microenvironment. However, in the studies reviewed here, such induction does not appear to play any role in the response, as most reviewed treatments result in decreased polyamine pools and would be expected to produce either no change in the expression of either or possibly even a decrease, since overall intracellular polyamine concentrations are reduced.
Reviewer 2 Report
see attached file

Author Response
My specific points are as follows.
- The authors state in the abstract “However, combination therapies targeting compensatory mechanisms have shown tolerability and efficacy in animal models and are currently being tested in clinical trials.” Maybe the authors could briefly mention currently ongoing trials in the main text, possibly including trial numbers? Are there any finished clinical trials of DFMO + PTI combination and what were the outcomes?
We thank the reviewer for this perspective and have made the following addition:
“Currently, there are four clinical trials listed (clinicaltrials.gov) investigating DFMO with chemotherapeutics in astrocytoma (NCT02796261, active) and neuroblastoma (NCT03794349, recruiting; NCT02030964, active; NCT01059071, completed). Additionally, one trial is investigating DFMO with the polyamine transport inhibitor AMXT 1501 in solid tumors (NCT03536728, recruiting), and another is investigating the polyamine analogue SBP-101 with gemcitabine in pancreatic cancer (NCT03412799, completed). Of note, only one trial has results posted (NCT01059071, Phase I) in which DFMO and etoposide co-treatment was well-tolerated and resulted in very low frequencies of serious adverse events.”
- Added to section 1 (revised lines 57-65)
- Lines 306 & 308. To me (as a non-expert in immunology) it was unclear what PD-L1, α-PD-L1, and α-PD-1 were (before checking from Pubmed). Maybe the authors could add abbreviation and possibly also a short description? The same with CSF1R (Line 379), CTLA-4 (lines 454-455), and JAML (line 481).
We acknowledge these clarifications and have made the following changes:
“immune-inhibiting programmed cell death-ligand 1 (PD-L1)”
- Added to section 3.2 (revised lines 313-314)
- We have added the phrase “immune-inhibiting” to specify that PD-L1 expression inhibits the immune response.
“immune-stimulating checkpoint inhibitor α-PD-L1 or α-PD-1”
- Added to section 3.2 (revised line 316)
- We have added the phrase “immune-stimulating” to specify that inhibition of the inhibitors PD-L1 and PD-1 will activate the immune response.
“anti-colony stimulating factor 1 receptor (α-CSF1R)”
- Added to section 3.4 (revised line 391)
- The following lines detail the consequences of blocking CSF1R and indicate its importance for macrophage polarization
“immune-inhibiting checkpoint cytotoxic T-lymphocyte associated protein 4 (CTLA-4)”
- Added to section 4 (revised lines 468-469)
- We have added the phrase “immune-inhibiting checkpoint” to specify that CTLA-4, like PD-1/PD-L1, inhibits the immune response.
“junctional adhesion molecule-like protein (JAML)”
- Added to section 4 (revised lines 498-499)
- We believe that further description is not necessary because JAML is only cited in our review in the context of a gene signature related to MDSCs [41].
- Lines 323-331. The results of the study REF #29 are a bit unclearly presented. It is missing the information that combination therapy with DFMO + PTI did reduce tumor growth. (But the anti- tumor effect of polyamine-blocking therapy was lost in mice where antibody-depleted CD4+ and CD8+ T )
We acknowledge this clarification and have made the following changes:
“subsequent”
- Added to section 3.3 (revised line 338)
- We have added this word to indicate that upon CD4 and CD8 depletion, subsequent DFMO and PTI treatment had no effect.
“Thus, cytotoxic and helper T cell populations may drive the anti-tumor response”
- Added to section 3.3 (lines 339-340)
- We have added this sentence to further clarify that the DFMO and PTI treatment-induced reduction of tumor burden is likely due to T cells.
- Lines 331-333. The sentence is incorrect. Actually, survival was measured only in Pan02 tumor model and survival of DFMO + PTI treated mice was significantly increased (Log-rank Mantel-Cox test, p < 0.0009). By contrast, combination therapy did not have additional benefit in reducing tumor weights in Pan02 and L3.6pl
The same for lines 393-395. Gitto et al. conclude “In summary, a combination therapy of DFMO + Trimer44NMe PTI was shown to nearly double the survival of tumor-bearing mice orthotopically injected with Pan02 cells.”
In addition, it would be good to include the used doses of PTI and DFMO for comparison of REF #30 and REF #31 results.
We thank the reviewer for this correction. To address these, we have made the following changes:
“Interestingly, while combined DFMO (1% w/v) and Trimer PTI (1.8 mg/kg) conferred significant survival benefit in the orthotopic Pan02 model of gemcitabine-resistant pancreatic cancer, it did not further reduce tumor burden when compared to DFMO or PTI alone [30]. Nakkina et al. increased the Trimer44NMe dosage to 4 mg/kg, (DFMO: 0.25% w/v) and observed a significant increase in survival, reduction in Pan02 pancreatic tumor burden, and increase in co-stimulatory marker CD86 expression, when compared to either treatment alone [31].”
- Added to section 3.3 (revised lines 340-348)
- Lines 334-336. Please add the used tumor model (REF #31).
“Pan02 pancreatic tumor burden”
- Added to section 3.3 (revised lines 346-347)
- Lines 340-342. It would be good to add whether the effect of DFMO + PTI was additive in comparison to either single
The authors do not state if the treatments were additive.
- Note that a similar review article was recently published https://pubmed.ncbi.nlm.nih.gov/35269518/
While there are similarities in the topics covered in the Chia et al. review, our review is more focused on presenting recent studies that aimed to decipher the impact of polyamine depletion strategies on the tumor immune microenvironment, and provides detailed methodologies engaged in their in vivo studies.
- Consider adding a paragraph about polyamine-blocking therapy in neuroblastoma. https://pubmed.ncbi.nlm.nih.gov/30700572/
https://pubmed.ncbi.nlm.nih.gov/19047152/
and other references.
Is there any data on modulation of immune system by polyamine-blocking therapy in neuroblastoma?
While these articles provide important findings in the context of neuroblastoma, they do not explore the implications of the polyamine depletion strategies on the tumor immune microenvironment, and thus we believe that these articles are beyond the scope of this review. We hope that future studies will generate these data to clarify the impact of PBT on immune-modulation in the context of neuroblastoma as well as other cancers, as the critical immune cell subtypes engaged in the anti-tumor response will likely differ depending on the tumor model system.
- The review deals with polyamine blocking therapy as anti-cancer approach. Has the effect of exogenous polyamine supplementation on tumor growth/survival been examined in any animal
model (or humans)? Is there association with increased polyamine intake from diet and tumor incident in humans or animal models?
Although polyamine supplementation studies are beyond the scope of the current review, there are significant amounts of polyamines in several foods and diets. No, specific, direct link between high polyamine diet and neoplasms has been described; however, reduced polyamine diet combined with polyamine depletion strategies have been explored in both animal models and humans.
- These articles may be of interest to the authors, about the role of SAT1:
We thank the reviewer for providing the links for the studies below. We are aware of those studies and found them very interesting.
https://pubmed.ncbi.nlm.nih.gov/24607957/
https://pubmed.ncbi.nlm.nih.gov/2383642/
Reviewer 3 Report
The manuscript entitled "Polyamine depletion strategies in cancer: remodeling the tumor 2 immune microenvironment to enhance anti-tumor responses" discusses the fact that although the polyamine synthesis-targeting drug 2-difluoromethylornithine (DFMO) is well tolerated in humans and is FDA-approved, its clinical benefit in treating established cancers has not yet been fully realized. It is shown that DFMO may sensitize tumors to other therapeutics including immunotherapies and chemotherapies.
The aim of this review is analysis of the immunomodulatory effects of polyamine depletion strategies.
The review is interesting and appealing to researchers and scientists.
It discusses new evidence that indicate how polyamines are critical for immune cell specification, and function. It also highlights that microbiota-derived polyamines are important for modulating macrophage function. It also discusses the immunomodulatory effect of DFMO in the context of polyamine blocking therapy.
The novelty relies in the diagram of polyamine metabolism and the polyamine blocking strategies which decrease immunosuppression in multiple animal models of cancer.
The paper is well written and the manuscript is well prepared.
Author Response
We thank the reviewer for their kind comments.